# A Non-Interventional Study Documenting Use and Success of Tissue Level Implants

**DOI:** 10.3390/ijerph17134816

**Published:** 2020-07-04

**Authors:** Carmen María Díaz-Castro, Pedro Lázaro Calvo, Francisco Javier Gil, Ana Fernández-Palacín, José-Vicente Ríos-Santos, Mariano Herrero-Climent

**Affiliations:** 1Department of Periodontics and Dental Implants, University of Seville, 41009 Sevilla, Spain; carmmaria@hotmail.com (C.M.D.-C.); dr.lazaro@madridperioimplant.com (P.L.C.); 2Bioengineering Institute of Technology, International University of Catalonia, 08017 Barcelona, Spain; xavier.gil@uic.cat; 3Department of Social and Health Sciences, Universidad de Sevilla, 41009 Sevilla, Spain; afp@us.es; 4Department of Periodontology, University of Seville, 41009 Sevilla, Spain; 5Porto Dental Institute, 4150-518 Porto, Portugal; dr.herrero@herrerocliment.com

**Keywords:** dental implants, immediate dental implant loading, bone–implant interface, osseointegration

## Abstract

Numerous randomised controlled multicentric studies have investigated various responses to different treatment modalities with dental implants. These studies do not always show the results of daily practice as they are performed under controlled and strict clinical conditions. This multicentric, non-interventionist trial aimed to document the behaviour of implants when used in daily dental practice, without inclusion or exclusion criteria. One hundred and ninety-six screw-shaped, tissue-level implants were placed, and each clinician decided which implant, surgical loading and prosthetic protocol to use. At surgery, data related to the implants were recorded. Additionally, the crestal bone level changes were evaluated for up to two years of follow-up. Two implants were lost before they were loaded. The success rate was 98.31%, and the survival rate was 98.79%. The implant stability quotient (ISQ) at surgery was 68.61 ± 10.35 and at 2 years was 74.39 ± 9.64. The crestal–shoulder distances were 1.25 ± 1.09 mm and 1.68 ± 1.07 mm in the mesial and distal aspects on the day of surgery, respectively, and 2.04 ± 0.91 and 2.16 ± 0.99 mm at 2 years, respectively. At 2 years, 69.3% of the patients were highly satisfied. The use of implants under standard conditions seemed to have success rates similar to their placement in controlled studies.

## 1. Introduction

Dental implants are, today, a predictable treatment for partially or fully edentulous patients and for single tooth replacements [1]. The literature shows that the survival rate of dental implants is greater than 95% over 8 to 10 years [2,3].

Primary implant stability (the absence of mobility in the bone site after implant insertion) is essential for adequate implant osseointegration [4,5]. Resonance Frequency Analysis (RFA) is an objective and non-invasive method for clinically testing the implant’s stability. Meredith introduced this technology in 1996, and it is amply demonstrated that is a useful device for impartially estimating the implant’s stability when the implant is placed or at any time during follow up. The literature shows that is a reliable and reproducible technology [6,7,8]. The RFA is performed by measuring the ISQ (implant stability quotient), ranging from 0 (no stability at all) to 100 (the highest value of stability that can be achieved) [5].

Crestal bone remodelling, the soft tissue dimensions (biologic width) and the degree of peri-implant inflammation, when implants with polished collars are placed, are influenced by where the rough/smooth interface (RSI) is located with respect to the bone crest and by the microgap between the prosthetic components and the machined collar [9,10,11,12,13,14]. As Cassetta et al. reported, in 2015, the position of the implant shoulder with respect to the bone crest influences marginal bone resorption, especially during the healing period [15]. 

Numerous articles have been published about the advantages and disadvantages of placing implants at different moments after tooth extraction. The different moments of placement, according to Hammerle 2004 [16], are type I (implant placement immediately after tooth extraction), type II (implant placement 4–8 weeks after tooth extraction), type III (implant placement 12–16 weeks after tooth extraction) and type IV (implant placement more than 16 weeks after tooth extraction). The literature shows that the survival rates of immediately placed implants have been similar to those of implants placed into healed bone and also similar to those of implants placed at early points in time [16,17]. 

Traditionally, two-stage surgery with a submerged healing period of 3 months for the mandible and 6 months for the maxilla was used to ensure osteointegration after implant placement. This technique has been verified in numerous clinical studies [18,19]. However, other trials have shown that non-submerged healing (one-stage surgery) could lead also to successful osseointegration and, at the same time, reduce the number of surgical interventions and discomfort [20,21,22].

With the increase in concern for aesthetics and interest in reducing treatment times, implants have begun to be loaded immediately after being placed. It has been demonstrated in different articles than implants could be loaded successfully immediately or early after their placement in selected patients. The most-used loading protocols, according to Esposito 2007 [23], are immediate loading (the implant is loaded before one week after placement), early loading (the implant is loaded in between one week and 2 months after placement) and conventional loading (the implant is loaded after 2 months of placement). Today, with the improvement of surface and implant characteristics, 2 months could be considered a conventional point in time at which to load implants. A recent clinical study demonstrated the effectiveness of this protocol in clinical practice [24].

Numerous randomised-controlled multicentric studies have investigated the survival rates, bone-level changes and various responses to the different treatment modalities with dental implants. These studies, although necessary, did not show the results for daily practice. That is because the trials were performed under controlled clinical conditions and in adherence to a precise protocol with strict inclusion/exclusion criteria.

The aim of this multicentric study is to document the behaviour of implants when used in daily dental practice, without inclusion or exclusion criteria. The products were used as they are in a typical dental office, but the usage was documented in a systematic way, and the results were analysed.

## 2. Materials and Methods

This was a multicentric, non-interventionist study, in which 12 dental centres (general and implant dentistry) participated. Each centre placed between six and 30 implants. The study began in January 2011 and finished in July 2014, including a recruitment period of 6 months. The follow-up period was 2 years.

The study was carried out in compliance with the Declaration of Helsinki (1964). The Ethics Committee at the University of Seville approved the trial, and all the patients gave written informed consent before the study commenced. The trial was monitored by the Master Program of Periodontology and Implant Dentistry of the University of Seville. The dentists who participated in the study were experienced clinicians (users and knowers of the Klockner Implant System for more than 5 years). The appropriate EQUATOR guidelines were complied with; the Declaration of Strengthening the Reports of Observational Studies in Epidemiology (STROBE) was followed—guidelines for reporting observational studies.

Before the study started, the clinicians participated in a calibration meeting to receive all the information (verbally and in writing) about the study and the assessment of the variables. A monitor was also appointed to answer questions during the study period. 

All the patients included in the study were over 18 years old and needed the placement of at least one implant to replace a tooth in the mandible or maxilla. Patients were excluded if they were pregnant, medically compromised (had metabolic diseases, had immunodeficiencies or received treatment with immunosupressive therapy, previously or currently used oral or intravenous bisphosphonates, received radiation therapy, etc.) or had any other local factor that could contraindicate implant surgery. Patients who participated in the trial were enrolled in the recruitment period to prevent them being withdrawn from the study if any complications arose. If a patient did not attend the study review appointments, the monitor attempted to locate them, to also prevent them from being excluded.

A full medical and dental history was registered, recording local (periodontal condition, poor oral hygiene, bruxism or bone defects) and systemic (e.g., diabetes, radiotherapy, physical disability, alcoholism or smoking) risk factors. Before surgery, photos, orthopantomography and a periapical X-ray of the edentulous areas could be taken, according to the criteria of the dentist who performed the treatment.

The implants employed in that study were screw-shaped, tissue level implants (Essential Cone, Klockner Implant System, SOADCO, Engordany, Andorra). The surface of these implants is called Shot Blasting and is a rough surface obtained by alumina-particle abrasion and acid passivation (Shot Blasting, Klockner Implant System). The implants used could have two different collar heights: 0.7 mm or 1.5 mm (Figure 1). The implant diameters used were 3.5, 4.0 and 4.8 mm, and the implant lengths used were 8, 10, 12 and 14 mm. The implant selection was made according to the criteria of the clinician.

As this was a non-interventionist study, each clinician decided which implant, surgical and prosthetic protocol—as well as the loading protocol—to use. At surgery, the following data were recorded:The implant position.The type of implant used.The moment of placement (according to Hammerle 2004 [16]).Quantity and bone quality (according to Lekholm y Zarb 1985 [25]): D1 (homogenous compact bone), D2 (thick layer of compact bone around a core of dense trabecular bone), D3 (thin layer of cortical bone around dense trabecular bone) and D4 (thin layer of cortical bone around a core of low-density trabecular bone).Primary stability (by means of insertion torque and frequency analysis resonance with the Osstell ISQ).The need for bone guided regeneration, if there was not enough bone to place the implant.The type of healing (submerged or non-submerged).The protocol for loading (according to Esposito 2007 [23]).The restorations made and the type of abutment, prosthesis material and type of retention (cemented or screw-retained) recorded for both the provisional and final prostheses.

The sutures were removed 15 days after the surgery, and 8 weeks after surgery, a control was done to evaluate the RFA, the mobility of the implant, if there was any complication and if the implant was successful according to Buser et al. (1990) [26] (the absence of persistent subjective complaints, absence of a recurrent peri-implant infection with suppuration, absence of mobility and absence of continuous radiolucency around the implant). The next control visits were at 6, 12 and 24 months; the following data were collected: the success and survival rate, RFA results, probing depth and bleeding on probing (in the mesial, distal, vestibular and lingual/palatal aspects of the implant) with the insertion of a standard periodontal probe with a point diameter of 0.5 mm using a probing force of 0.5 N, modified plaque index (Mombelli et al. 1987 [27]), and satisfaction degree of the patient [28].

The RFA was measured with the Osstell ISQ following the manufacturer’s instructions. The transducers were screwed using a specific hand-screwdriver (approximately 6–8 Ncm of torque) to the implant, and the ISQ was recorded (perpendicular to the transducer, approximately 2 mm from it, in a vestibular–lingual direction).

The crestal bone level changes were evaluated by measuring the shoulder–crest distance (SCD) on the mesial and distal sides of each implant by means of periapical customised radiographs. The X-rays were taken with the parallel technique, customising the film holder (Rinn XCP, Dentsply Rinn) by a bite registration with a polysoloxane material (Normosil Addiction Putty Normal) on the bite block and resting on the opposite arcade [14]. X-rays were taken on Day 0 (the day when the implants were placed), at 8 weeks and at 6, 12 and 24 months after implant placement. 

The SPSS 14.0 software (SPSS, Chicago, IL, USA) was used for the statistical analysis. Mean values and standard deviations were calculated for quantitative variables, and percentages, for qualitative ones. The normal distribution of the values and the homogeneity of the variances were tested through a Kolmogorov–Smirnov test. The differences between the mean values were compared with Student’s t test for quantitative variables. When significant differences were observed, the 95% confidence intervals were determined for the average and mean differences (*p* < 0.05). For qualitative variables, the chi-squared test was used.

## 3. Results

A total of 92 patients (35 men and 57 women) participated in the study, in which 196 implants were placed. The mean age of the participants was 48.6 years at the time of implant placement. Of the implants, 76% were placed in non-smoking patients; 9.2%, in patients who smoked fewer than 10 cigarettes a day; and 13.3% of the implants were placed in smokers of more than 10 cigarettes a day.

### 3.1. Implants Used

The diameters of the implants used were 3.5 mm (50 implants), 4 mm (101 implants) and 4.8 mm (45 implants). Implants of 8 (18 implants), 10 (83 implants), 12 (79 implants) and 14 mm (16 implants) in length were placed. Of the 196 implants, 80 were implants with machined collars of 0.7 mm, and 116, with machined collars of 1.5 mm. Of the implants, 44.9% were placed in the upper jaw, and the other 45.1% were placed in the lower jaw. The locations of the implants by tooth site (FDI nomenclature) are displayed in Figure 1. The most frequent location was the first lower molar. 

According to the moment of implant placement (Hammerle 2004), 15 implants were type I, 10 were type II, 13 were type III and 158 were type IV implants. Eight implants were placed in bone type I, 110 were placed in bone type II, and 64 and 14 implants were placed in bone types III and IV, respectively.

As it was a post-marketing study of consecutive cases, not all the centres recorded all the variables for different reasons (X-rays were not appropriate for measurements, or some prostheses—cemented ones—could not be removed to measure the ISQ values, so from this point on, when not all the implants were analysed, the number of them is indicated.

At the moment of placement, 186 implants had primary stability. The ISQ on the day of the surgery was 68.61 ± 10.35; at 8 weeks, it was 70.61 ± 7.5; and at 2 years (when only the ISQs of 61 implants were analysed), it was 74.39 ± 9.64. How the implants healed was registered in 129 implants: 20.9% of the implants healed in a submerged way; 42.6%, in a semi-submerged way; and 36.4%, in a non-submerged way.

### 3.2. Drop Outs

During the study, there were some drops out because of illness and personal motives: two patients left the study after the surgery; one patient left the study after the revision at 8 weeks; one patient give up the study after the placement of the definitive prosthesis; two patients left the study without completing the 6, 12 and 24 month visits; one patient left the study after the 6 month visit; and four patients did not attend the 24 month visit. Some patients missed intermediate appointments: one patient did not attend the 6- and 12-month visits, one patient left the 6 month visit, one patient missed the 12 month visit, one patient did not attend the appointment for prosthesis placement and the 6 month visit, and one patient missed the prosthesis placement and 24 month visits.

### 3.3. Survival and Success Rates and Complications

Two implants were lost before they were loaded. The success rate was 98.31%, and the survival rate was 98.79% (Table 1). Some prosthetic complications (in 5.4% of the patients) occurred during the study period time: one abutment was broken, two implants suffered from screw loosening, two prostheses were changed to improve the aesthetics, one crown was changed from cement-retained to screw-retained, one implant presented a gap with the crown and one implant-crown had a ceramic fracture.

### 3.4. Periodontal Health

The periodontal health (described by the plaque index, bleeding on probing and the probing depth criteria) of the implants is shown in Figure 2, Figure 3 and Figure 4. At 8 weeks, 183 implants were analysed for the plaque index and probing depth, and 181, for the bleeding on probing. At 6 months, 174 implants were analysed for the plaque index; 176, for the probing depth; and 167, for the bleeding on probing. At one and two years, 165 implants were analysed for the plaque index and probing depth, and 166, for the bleeding on probing.

### 3.5. Crestal–Shoulder Distance (SCD)

The crestal–shoulder distances (SCDs) were 1.25 ± 1.09 mm and 1.68 ± 1.07 mm in the mesial and distal aspects, respectively, on the day of the surgery, and 2.04 ± 0.91 and 2.16 ± 0.99 mm, respectively, at 2 years (when the X-rays of 119 implants were examined). The SCDs according to the type of the collar and the moment of implant placement are displayed in Figure 5 and Figure 6, respectively. For the analysis of these data, 139 X-rays were studied at the moment of surgery; 126, at 8 weeks; 78, when the definitive prostheses were placed; and 111, 115 and 119, at 6 months, 1 year and 2 years, respectively. 

### 3.6. Restorations

Of the implants, 98.9% were restored with metal–ceramic prostheses, and just 1.1% of the implants were restored with resin prostheses. A proportion of 56.8% of the implants were rehabilitated with screw-retained prosthesis, 34.1% of the implants were restored by single crowns, 64.8% were restored by partial fixed prostheses, and 1.1% were restored by overdentures. For the analysis of those data, 176 implants were studied. 

### 3.7. Comfort and Satisfaction

At one year (when 164 implants were analysed), 59.1% and 47.5% of the patients scored the comfort and the aesthetics of the prostheses as good, respectively; 58.5% of the patients were highly satisfied. At 2 years (when 166 implants were analysed), 68.7% and 55.4% of the patients scored the comfort and the aesthetics of the prosthesis as good, respectively; 69.3% of the patients were highly satisfied.

## 4. Discussion

The present study tried to reflect the real employment of implants in dental practice. The success and survival rates obtained in this study are similar to the ones observed in controlled clinical trials [14,29]. The systematic review of Moraschini et al. in 2015 [30] found a mean survival rate of 94.6% (±5.97%) for a total of 7711 implants, with a follow-up period of up to 20 years (mean follow-up of 13.4 years). This systematic review also demonstrated that the survival rate decreases as the follow-up period increases. The results are also in agreement with those of no controlled trials. Cacaci et al. [31], found a survival rate of 98.4% at 36 months. and Beschnidt et al. [32], in 2018, found cumulative survival rates of 100% at 1-year follow-up, 99.6% at 3-year follow-up and 98.6% at 5-year follow-up.

Controlled clinical trials, although necessary, do not always show results based on ordinary circumstances. In the development of these studies, implants are placed under special circumstances: the investigators have more time to treat patients (following strict protocols), the patients selected meet strict exclusion and inclusion criteria (related to tobacco habits, systemic health and dimensions of the bone for implant placement). 

In the present study, different treatment protocols were applied: different types of implantation (different moments of placement), healing procedures (submerged or non-submerged), and prosthetic restorations (screw- versus cement-retained, single crowns and Fixed Partial Dentures). As in numerous trials and systematic reviews, high survival and success rates as well as excellent clinical conditions without statistically significant differences between the applied treatment types have been shown [17,31,33,34]. The study results confirm that the surgical and restorative techniques had almost no influence on the outcomes.

Primary implant stability is considered a crucial requirement for adequate osseointegration. The maintenance of appropriate stability over time is also considered a long-term guarantee of success [5,35]. RFA allows the objective quantification of implant stability at any given time, as well as implant monitoring, allowing sequential measurements and the study of implant behavior over time [36,37]. ISQ values over 60 are considered indicators of appropriate implant stability. The primary stability values obtained in this study are higher than those reported by other authors such as Zix et al. [38], whose average ISQ values were 52.5 ± 7.9, or Östman et al. [39], whose average stability value was 67.4 ± 8.6. The reason for these differences could be due to the implant design, surgery technique or bone density.

There was an increase in the ISQ values from implant placement to 2 years after implant placement. This increase in stability would be due to bone formation and maturation around the implants, as well as the healing process of bone tissue in the implant surface. The work by Barewal et al. [40] reported the results of weekly RFA up to the 10th week, proving there were no significant changes in implant stability after the fifth week. A similar registration pattern was found by Boronat et al. [41], whose study reported the weekly stability values for 64 implants placed in 24 patients with deferred loads. The highest values were found in the 10th week, and the lowest ones, in the fourth week (60.9 ISQ). The average ISQ value was 62.6. 

Most of the implants were placed in bone types II and III. This distribution is in accordance with that described in other non-interventionist studies such as the study of Cacaci et al. [31], in which 51.5% and the 33.1% of the implants were placed in bone types II and III, respectively, or that of Beschnidt et al. [32], in which 42.2% of the implants were placed in bone type II and another 42.2%, in bone type III.

The probing pocket depth during all study period was 4 mm or less in over the 95% of the implants, which is in agreement with the norm for conventionally placed implants, which at 2–4 mm is indicative of healthy tissue [32,42]. The bleeding on probing remained very low throughout the study, despite a significant increase occurring from 1 year after implant placement. However, the plaque index was very similar during the whole study.

One of the criteria for evaluating the success of the implants was the stability of the marginal bone level [43]. The marginal bone level was usually evaluated by means of the SCD. The SCD in which the biological width is established depends on several factors such as the location of the implant–abutment gap; the height of the polished collar, if it exists; the gap–bone crest distance; the relationship between the rough/smooth interface (RSI) and bone crest; and the design of the implant–abutment junction. Bone resorption would occur during implant healing, to create the necessary space for the connective tissue adaptation [14,44,45].

The results of the present study showed that the SCDs of the implants with the 0.7 mm machined collar neck were greater at 2 years than the SCDs of those with 1.5 mm machined collars. However, on the day of surgery, the SCD for the implants with the 0.7 mm machined collar neck was 0.92 mm, and that for those with the 1.5 mm ones was 1.57 mm. That means that the implants with the 0.7 mm machined collar necks were placed slightly with the RIS subcrestally. The changes between surgery and 2 years that took place in the 1.5 mm machined collar (0.42 mm) were less than those reported in the literature [46]. The literature reported that implants placed subcrestally exhibit the greatest mean bone loss compared to those in which the RIS is located at the level of the bone crest [14,47,48,49,50]. 

When analysing the SCD according to the moment at which the implant was placed, the greatest SCD at 2 years was that for the type IV implants, and it was so at the moment of implant placement. The lowest SCDs were those for the immediate implants at both baseline and 2 years, which means that those implants were placed with part of the RIS subcrestally, which is in concordance with the technique of immediate implant placement. 

From the data of the study, the bone loss could be calculated as follows: 0.85 mm, 0.15 mm, 0.22 mm and 0.65 mm for the type I, II, III and IV implants, according to Hammerle et al. The higher bone loss in the immediate implants shown in this study is not in agreement with the results of other studies such as the meta-analyses published by Kinaia et al., where the immediate implants showed less bone loss [51]. Additionally, Grandi et al. [52] showed less bone loss in immediate implants than in early or conventional placed implants, although the differences were not statistically significant. The differences with the former study’s results could be due to Grandi et al. employed platform-switched implants. Patattella et al. [53] compared type I and type II implants and observed that the bone loss after 2 years was 0.54 ± 0.51 mm in the immediate group and 0.46 ± 0.54 mm in the early group, with the difference not being statistically significant. These results follow the trend of this study, in which the early-placed implants are also the ones with the lowest bone loss. The controversial findings in the literature about the bone loss with respect to the moment of implant placement could be due to different factors such as flap or flapless surgeries, immediate or non-immediate loading or the lack thereof, and regeneration procedures or the lack thereof. The studies mixed all these factors and could be not strictly compared.

One of the limitations of the trial was that the clinician selected which implants to include in the study, so it is possible that the more challenging cases were not selected to be included. However, the external validity of the trial could be greater than that of controlled clinical trials because the participants went to non-university clinics and the inclusion criteria applied were not as strict as those when implants are placed for that kind of study. 

Another limitation of the study could be the drop-out rate, as 12% of the patients left the study or missed an appointment. 

## 5. Conclusions

In conclusion, implants placed under standard conditions seemed to have success rates similar to those placed in controlled studies.

## Figures and Tables

**Figure 1 ijerph-17-04816-f001:**
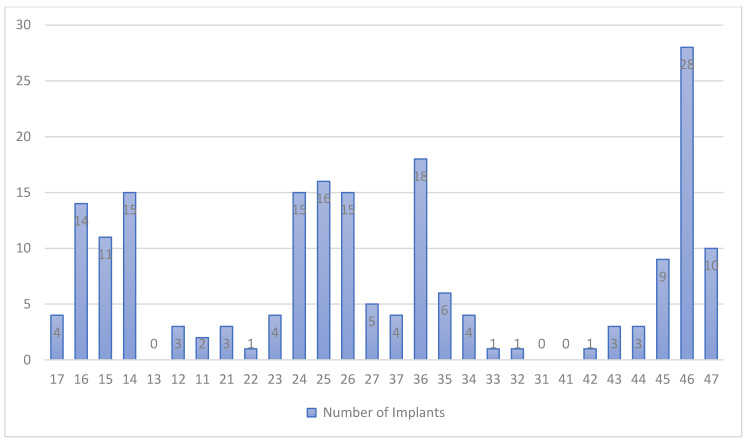
Location of implants.

**Figure 2 ijerph-17-04816-f002:**
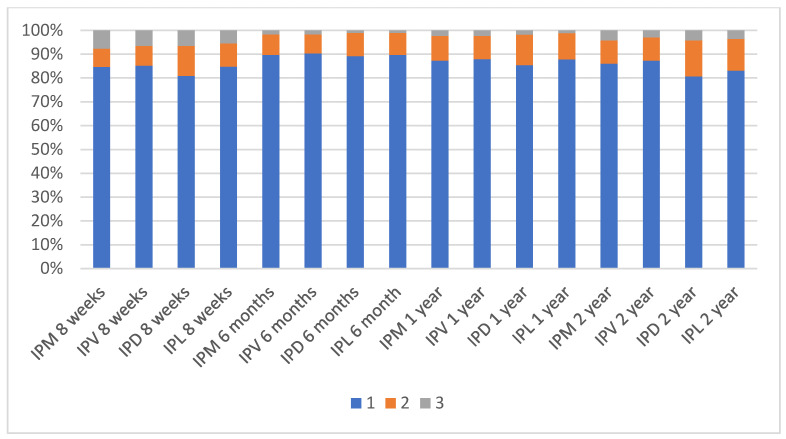
Plaque índex (Mombelli et al. 1987).

**Figure 3 ijerph-17-04816-f003:**
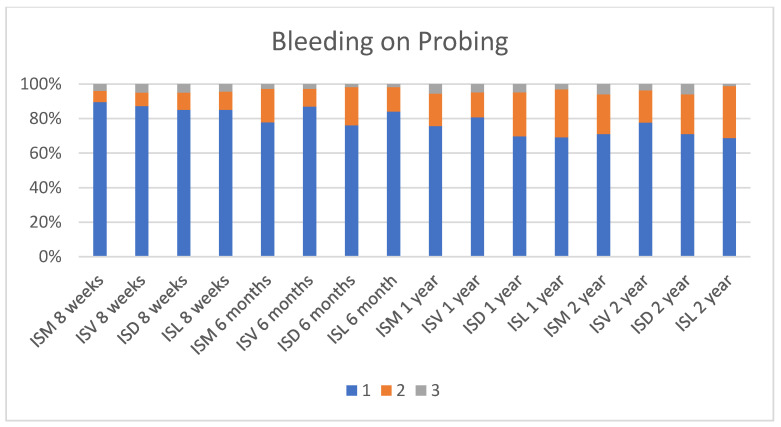
Bleeding on probing (Mombelli et al. 1987).

**Figure 4 ijerph-17-04816-f004:**
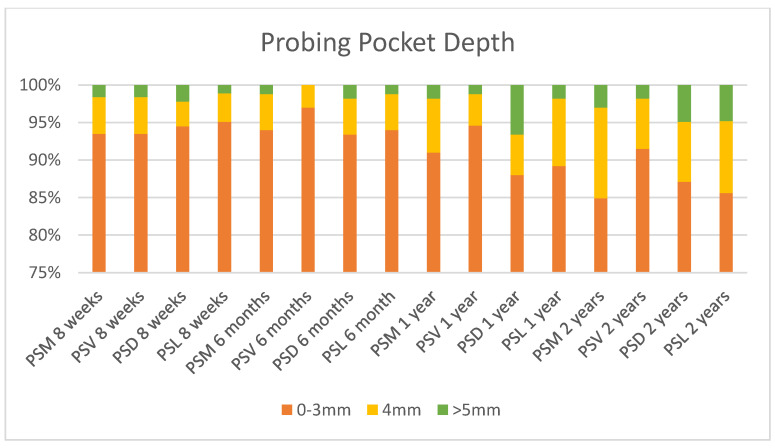
Probing pocket depth.

**Figure 5 ijerph-17-04816-f005:**
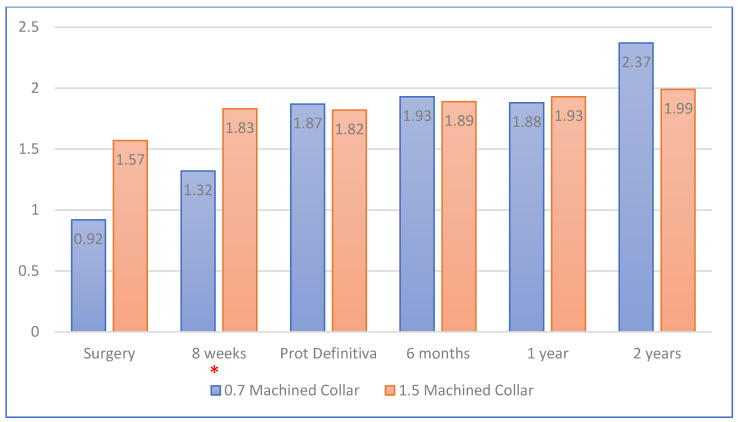
Shoulder–crest distance (SCD) (in mm) according to the machined collar of the implant. * Statistically significant differences (*p* < 0.05).

**Figure 6 ijerph-17-04816-f006:**
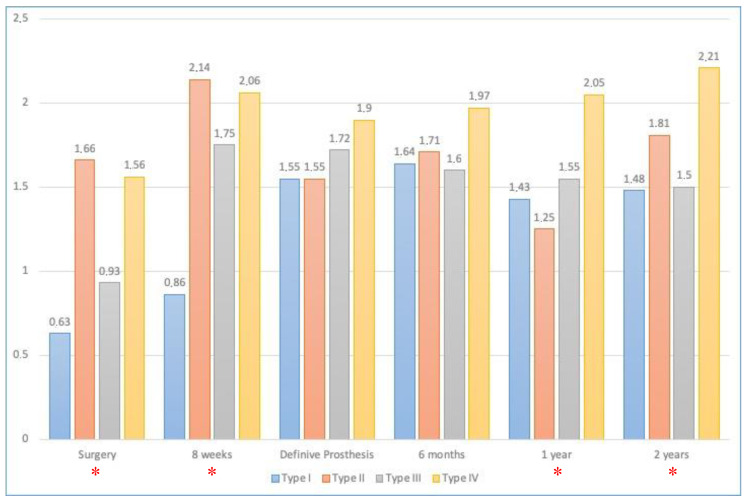
SCD (in mm) according to the moment of implant placement. * Statistically significant differences (*p* < 0.05).

**Table 1 ijerph-17-04816-t001:** Survival (implants that were present at the moment of examination) and success (implants that met the Buser criteria [26]) rates. The rates were calculated according to the number of implants analysed at each moment.

Time	Implants Failures	Survival		Success	
	2	**No of Implants**	**Survival Rate**	**No of Implants**	**Success Rate**
6 months		176	98.86%	111	98.15%
12 months		166	98.79%	115	98.20%
24 months		166	98.79%	119	98.31%

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
