# Peer review of "A Non-Interventional Study Documenting Use and Success of Tissue Level Implants"

_ijerph, 2020, doi:10.3390/ijerph17134816_

Round 1

Reviewer 1 Report

Regarding the summary I think it would be clearer if it included the fact that there are 196 implants with a two-year follow-up period

Introduction

Line 73: The study does not really test ’The null hypothesis is that the behavior used in daily practice is the same of those used in controlled clinical trials” however, this analysis is not the object of the study.

The study is a multicentric, descriptive study that provides information on the usual clinical behaviour of screw-shape, tissue- level implants.  The study analyses the crest-shoulder distance according to the machined collar of the implant and according to the time of implant placement, and also recorded information regarding comfort and satisfaction.

Material and method

It would be interesting to know how the patients were recruited, and although the discussion states it was not particularly restrictive, the criteria used should be reported

Line 106 They forgot to include they also used 14 mm implants as mentioned in the results

 Results

Table 1 survival and success table, should explain correctly where each datum comes from likewise how survival and success rates were calculated.

Discussion
Taking into account 12% of the patients were lost, (we know nothing  about the implants) it would be a good idea to have a comment on the matter in the discussion.

Author Response

Comments and Suggestions for Authors

Regarding the summary I think it would be clearer if it included the fact that there are 196 implants with a two-year follow-up period

We have added this to aspects to the summary.

Introduction

Line 73: The study does not really test ’The null hypothesis is that the behavior used in daily practice is the same of those used in controlled clinical trials” however, this analysis is not the object of the study.

According to your suggestions, this sentence has been eliminated.

The study is a multicentric, descriptive study that provides information on the usual clinical behavior of screw-shape, tissue- level implants. The study analyses the crest-shoulder distance according to the machined collar of the implant and according to the time of implant placement, and also recorded information regarding comfort and satisfaction.

Material and method

It would be interesting to know how the patients were recruited, and although the discussion states it was not particularly restrictive, the criteria used should be reported

As you suggest, we have tried to explain better the inclusion and exclusion criteria.

Line 106 They forgot to include they also used 14 mm implants as mentioned in the results

Implants of 14mm were added.

Results

Table 1 survival and success table, should explain correctly where each datum comes from likewise how survival and success rates were calculated.

According to your suggestions, it was explained in legend of table 1.

Discussion

Taking into account 12% of the patients were lost, (we know nothing about the implants) it would be a good idea to have a comment on the matter in the discussion.

We have included a reference to it at the end of the discussion section.

Reviewer 2 Report

Page 1 line 22: The use of the present tense is not appropriate.

Page 2 line 45: The position of the implant shoulder with respect to the bone crest influences marginal bone resorption especially during the healing period. Please add this concept and cite : “Cassetta M, Pranno N, Calasso S, Di Mambro A, Giansanti M. Early peri-implant bone loss: a prospective cohort study. Int J Oral Maxillofac Surg. 2015 ;44:1138-45.”.

Page 2 line 67: Please specify that the conventional load today is two months. There are recent clinical studies that demonstrate the effectiveness of this protocol in clinical practice. Please cite ”Cassetta M., Brandetti G., Altieri F.Is a Two-Month Healing Period Long Enough to Achieve Osseointegration? A Prospective Clinical Cohort Study Int J Oral Maxillofac Surg 2020;49:649-654”.

Page 2, line 79: Each centre placed in between six and 30 implants, distributed as follows: 11,12, 30, 19, 14, 24, 14, 11, 16, 19, 20, 6 implants. : this information can be omitted.

Page 3 line 112, please correct typo.

Author Response

Comments and Suggestions for Authors

Page 1 line 22: The use of the present tense is not appropriate.

The present tense was changed.

Page 2 line 45: The position of the implant shoulder with respect to the bone crest influences marginal bone resorption especially during the healing period. Please add this concept and cite : “Cassetta M, Pranno N, Calasso S, Di Mambro A, Giansanti M. Early peri-implant bone loss: a prospective cohort study. Int J Oral Maxillofac Surg. 2015 ;44:1138-45.”.

It was added, as you suggested.

Page 2 line 67: Please specify that the conventional load today is two months. There are recent clinical studies that demonstrate the effectiveness of this protocol in clinical practice. Please cite ”Cassetta M., Brandetti G., Altieri F.Is a Two-Month Healing Period Long Enough to Achieve Osseointegration? A Prospective Clinical Cohort Study Int J Oral Maxillofac Surg 2020;49:649-654”.

It was added, as you suggested

Page 2, line 79: Each centre placed in between six and 30 implants, distributed as follows: 11,12, 30, 19, 14, 24, 14, 11, 16, 19, 20, 6 implants. : this information can be omitted.

We have omitted the number of implants place in each center, as you asked for.

Page 3 line 112, please correct typo.

It has been corrected.

Reviewer 3 Report

This is a work on the implant success rate in different clinical structures
Although, in itself, the topic would be interesting in my opinion, many critical aspects of the study remain which do not make it suitable for its publication in the magazine.
The design of a study must include strict inclusion and exclusion criteria. In fact, implant success cannot disregard these aspects, as well as the clinical skills of surgeons.

Author Response

Comments and Suggestions for Authors

This is a work on the implant success rate in different clinical structures

Although, in itself, the topic would be interesting in my opinion, many critical aspects of the study remain which do not make it suitable for its publication in the magazine.

The design of a study must include strict inclusion and exclusion criteria. In fact, implant success cannot disregard these aspects, as well as the clinical skills of surgeons.

We appreciate your opinion about our work. As our aim is to test the behavior of implants in the daily clinic practice, the inclusion and exclusion criteria are no too strict. Anyway, we have tried to explain better that issue in our paper.

Round 2

Reviewer 3 Report

as already expressed in the previous review, despite the improvements made by the authors, I consider the manuscript not suitable for the scientific journal and I recommend rejecting it

Author Response

sent to the Academic Editor